# Sclerochronological characteristics of *Orbicella faveolata* in Cayo Arenas, a remote coral reef from the Gulf of Mexico

**D. Wendoline Sánchez-Pelcastre[1], J. J. Adolfo Tortolero-Langarica[1,2], Lorenzo Alvarez-Filip [3], Israel Cruz-Ortega[1], Juan P. Carricart-Ganivet[1] ***

**1** Laboratorio de Esclerocronología, Unidad Académica de Sistemas Arrecifales, Instituto de Ciencias del Mar y Limnología, Universidad Nacional Autónoma de México, Puerto Morelos, Quintana Roo, México,
**2** Tecnológico Nacional de México/ IT Bahía de Banderas, Bahía de Banderas, Nayarit, México,
**3** Biodiversity and Reef Conservation (BARCO) Laboratory, Unidad Académica de Sistemas Arrecifales, Instituto de Ciencias del Mar y Limnología, Universidad Nacional Autónoma de México, Puerto Morelos, Quintana Roo, México

\* carricart@cmarl.unam.mx

**Data Availability Statement:** All relevant data are in supplementary material as Appendix 1.

## Abstract

During coral calcification in massive scleractinian corals, a double annual banding of different densities (high- and low-density) is formed in their skeletons, which can provide a retrospective record of growth and the influence of environmental conditions on the coral's lifespan. Evidence indicates that during the last decades, the reduction in coral calcification rate is attributed to the combination of global stress factors such as Sea Surface Temperature (SST) and local anthropic stressors. Yet, coral growth trajectories can vary between regions and coral species, where remote locations of coral reefs can act as natural laboratories, as they are far from the harmful effects of direct anthropogenic stressors. The present study reports historical chronology over a 24-year period (1992–2016) of coral extension rate (cm yr⁻¹), skeletal density (g cm⁻³), and calcification rate (g cm⁻² yr⁻¹) of the reef-building coral *Orbicella faveolata* at the remote reef Cayo Arenas, Campeche Bank, in the southeastern Gulf of Mexico. The relationships between the three sclerochronological features show that *O. faveolata* uses its calcification resources to build denser skeletons. Chronological trends indicate that coral extension increased, skeletal density and calcification rate decreased (33% calcification rate) over time. The results reveal that despite the remoteness of the locality the maximum SST has been increased, and the coral calcification rate decreased over time. If the temperature continues to rise, there is a conceivable risk of experiencing a decline in reef-building coral species. This scenario, in turn, could pose a significant threat, endangering not only the framework of coral reefs but also their ecological functionality, even within remote Atlantic reef ecosystems.

## Introduction

Environmental reconstructions can be obtained on living organisms such as scleractinian corals, which may provide crucial information on the natural response and acclimatization/

**Funding:** JPCG UNAM-DGAPA-PAPIIT IN200420 LAF UNAM-DGAPA-PAPIIT IG201323 The funders had no role in study design, data collection and analysis, decision to publish, or preparation of the manuscript.

**Competing interests:** The authors have declared that no competing interests exist.

adaptation of species, leading to a better understanding and application of management and conservation measures in ecosystems to future climate scenarios [1, 2]. Growth in reef-building corals results from the accumulation of large amounts of calcium carbonate ($CaCO_3$) in their skeletons. This is known as coral calcification and it is the leading process that builds and maintains both the physical framework and the balance ecological functionality of the reef ecosystems [3, 4].

Coral calcification is mediated by environmental factors such as light irradiation, water temperature, water chemistry, nutrient concentration, and others [5]. However, the seawater temperature is one of the most important variables that control the variation in calcification rates and skeletal growth of scleractinian corals, and it may determine their distribution along spatial gradients [5–8]. However, prolonged exposure to a temperature that is +1˚C above the species threshold can cause thermal stress to coral organisms, leading to the expulsion of the endosymbiotic algae (*Symbiodinium*) in an event known as coral bleaching. If the stress persists for more than four weeks, corals may lose the capacity to maintain vital physiological process and may even perish [9–11]. In the context of rapidly changing climatic conditions, temperature increases have become one of the most important threats to coral growth, calcification, and survival [9].

During the process of coral calcification and skeletal growth, corals form high- and low-density bands of calcium carbonate (g $cm^{-3}$) between the summer and winter seasons, respectively [11–13]. These bands act as natural "sclerochronometers" similar to tree rings in dendrochronology [12, 14], allowing the history of tropical ocean and coral reef ecosystems to be traced through the calcium carbonate skeletons of reef-building corals [11, 15]. By analyzing the annual rhythmic patterns in the coral skeleton, we can reconstruct the coral's life history and understand how these organisms respond to environmental factors through coral growth characteristics such as skeletal density (g $cm^{-3}$), extension rate (cm $yr^{-1}$), and calcification rate (g $cm^{-2}$ $yr^{-1}$) rate [11, 15, 16]. Sclerocronology is an important tool to understand the effects of climate change on coral calcification rates and the historical response of coral reef species [11, 12, 17, 18]. Furthermore, the study of coral growth can be used to understand past climate events, such as thermal anomalies, pH, and nutrient concentration, through the analysis of atypical high-density bands known as "stress bands," which result from the sensitivity response of corals species to environmental variability [17].

Over the past twenty years, many studies have indicated a reduction of 11–21% in calcification rates within the tropical regions of the Great Barrier Reef located in Australia [9] Nonetheless, considering their life cycles and adaptation mechanisms, individual species exhibit distinct growth rates influenced by local and regional environmental factors [5, 18, 19]. It is plausible that corals might adapt differently across many reef regions. In the west-Atlantic and Caribbean region, the massive reef-building coral *Orbicella* spp. significantly contribute to the formation and maintenance of coral reef ecosystems [20]. Because of its morphological characteristics and its abundance, this species is one of the coral species most used for sclerochronological reconstruction in the region. Yet, most of these reports have been conducted in reef locations or periods influenced by both natural and local anthropic factors, which makes it difficult to identify the level of influence of each factor [7, 8, 21–24]. The local anthropogenic effects are important, as in addition to temperature, coral growth is influenced by the variability of environmental factors such as nutrient load [25, 26], sedimentation [21], depth [27], wave exposure [28]. Therefore, estimating calcification and growth rates in remote locations isolated from local human influence would allow us to clearly see the effects of climate change.

Cayo Arenas in the Campeche Bank, Gulf of Mexico is a small reef bank located ~170 km from the coast north of the Yucatan Peninsula. This reef can be considered a "natural laboratory" due to its remoteness, this site presents the possibility of studying the natural dynamics

of biological and ecological aspects far from local anthropogenic pressures, as well as assessing the direct effects of climate change on coral reefs. The objective of this study was to establish long-term baselines of growth (skeletal density, extension rate, and calcification rate) of *O. faveolata* from Cayo Arenas, by evaluating historical coral growth trends from 1992 to 2016, and the effect of seawater temperature on calcification rates. These results underline the need to improve our understanding of the role that temperature plays in the growth of scleractinian corals in the Atlantic region, considering that the increase in ocean temperature and global climate changes may exacerbate coral reef degradation [29–31].

## Materials and methods

### Study area

Campeche Bank is an insular area with a series of small emerging coral reefs and submerged coral banks in shallow waters, scattered along the western edge of the southwestern Gulf of Mexico, at 150–200 km from the coast of the Yucatan Peninsula, Mexico (Fig 1) [32, 33]. The dynamic of the ocean waters in the Cayo Arenas reef is determined by the Yucatan current (Loop Current), which infiltrates from the east to the west of the Gulf of Mexico, causing cyclonic and anticyclonic circulation zones [34] with an intra-annual SST, from 29.1°C during June to September and colder waters 23.5°C from January to April [35]. The Cayo Arenas reef systems comprise a group of three distributed emergent reefs spread over an area of ~5 km$^2$. It has a well-developed coral structure [35], where the coral community is mainly composed of *Montastrea cavernosa*, *Orbicella annularis*, *O. faveolata*, *Acropora palmata*, *Pseudodiploria strigosa*, *Siderea siderastrea* (Fig 2) [36].

### Coral sampling and analysis of sclerochronological variables

Coral samples of massive *Orbicella faveolata* were collected (n = 5 cores) in August 2017, at 7–12 m depth. Coral cores were extracted from the maximum apical growth axis of each colony, using a submersible drill (Nemo Divers Edit 50M) and a diamond-tipped core bit with a diameter of 5 cm and a length of 20 cm (coral cores length ranged between 14 to 19.5 cm). After obtaining the samples, a styrofoam ball was placed to prevent the invasion of boring organisms and help the recovery of live coral tissue [37]. At the laboratory, all samples were cut longitudinally with a diamond-edged saw into slabs of ~7 mm thick and then dried using a conventional oven at 70°C for 24 hours. Coral slabs were radiographed with conventional X-ray equipment (GE HungayRt Medical Systems) at a 1.80 m length between the ray source and coral slabs. To establish skeletal density patterns, a density standard wedge of the giant clam *Tridacna maxima* of known density (2.82 g cm$^{-3}$) was used in the radiographic process [38].

The radiographs were digitized into 75 dpi images (Kodak Direct View Classic Cr System) and corrected from errors associated with the heterogeneity in the irradiance bias received by the X-rays using the technique proposed by Duprey et al. [39]. From the corrected images we analysed them using the software ImageJ ver. 1.45, to obtain a data series of absolute density and extension following tracks along the main growth axis using the optical densitometry method described by Carricart-Ganivet and Barnes [38]. From these series, three sclerochronological characteristics were obtained: (1) the average annual skeletal density (g cm$^{-3}$) defined as the mean density value between maximum adjacent density peaks, (2) the annual extension rate (cm yr$^{-1}$) obtained by measuring the distance between maximum density peaks, and (3) annual calcification rate (g cm$^2$ yr$^{-1}$) calculated as the product of the mean density and extension rate for the corresponding year (Fig 3) [38, 40].

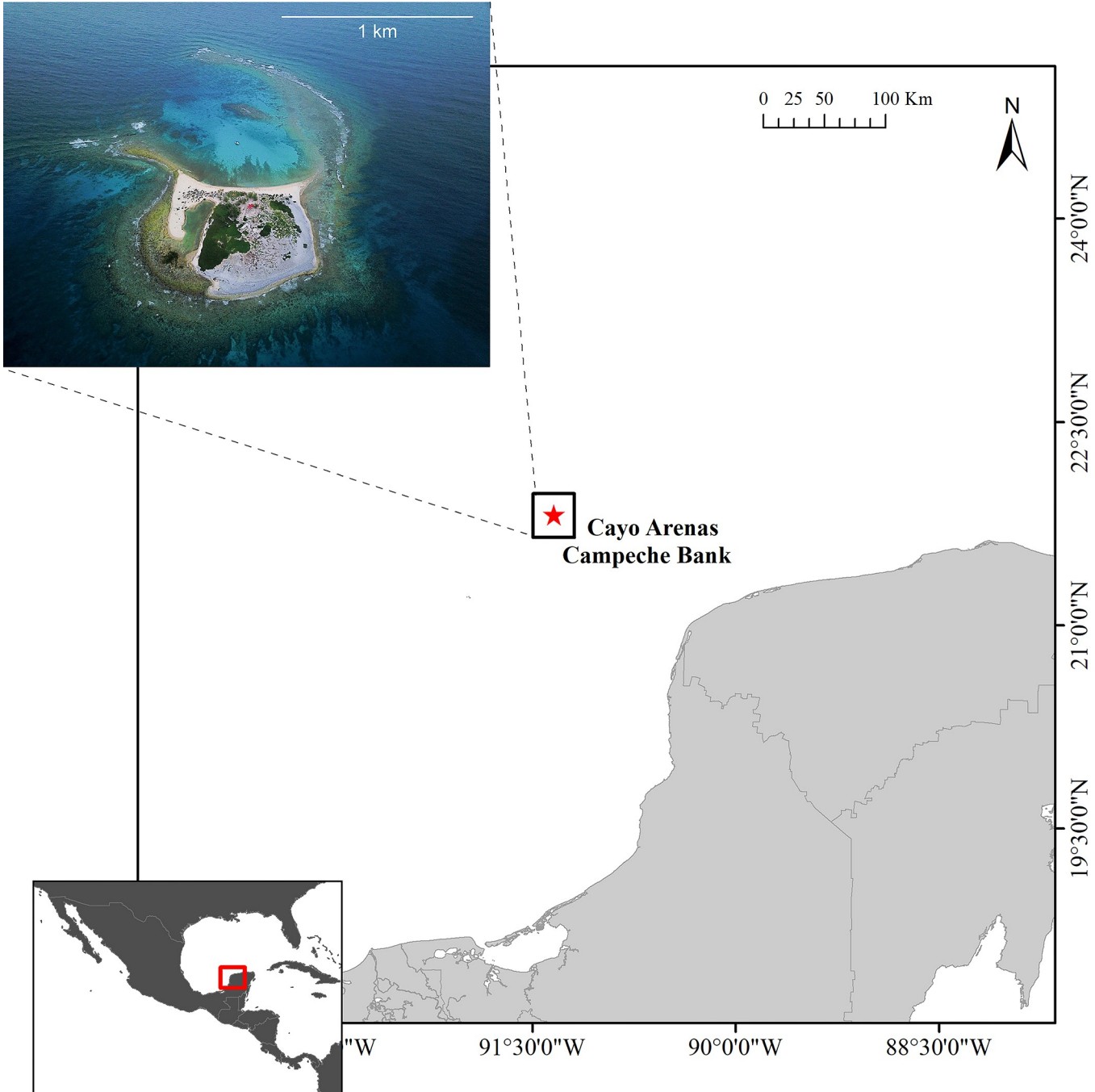

**Fig 1. Location of the reef where corals were collected.** Cayo Arenas reef, Campeche Bank in the Gulf of Mexico. Maps created by J.J. Adolfo Tortolero-Langarica and the aerial image of the island was taken by drone by Lorenzo Álvarez-Filip.

### Environmental data acquisition

The sea surface temperature (SST) was used as an environmental metric. Monthly SST dataset from 1992 to 2016 of Campeche Bank was obtained from satellite images (1° latitude-longitude grid resolution) acquired from the Hadley Centre Sea Ice and SST (HadISST, https://www.metoffice.gov.uk/hadobs/), dataset produced by the United Kingdom Meteorological Office

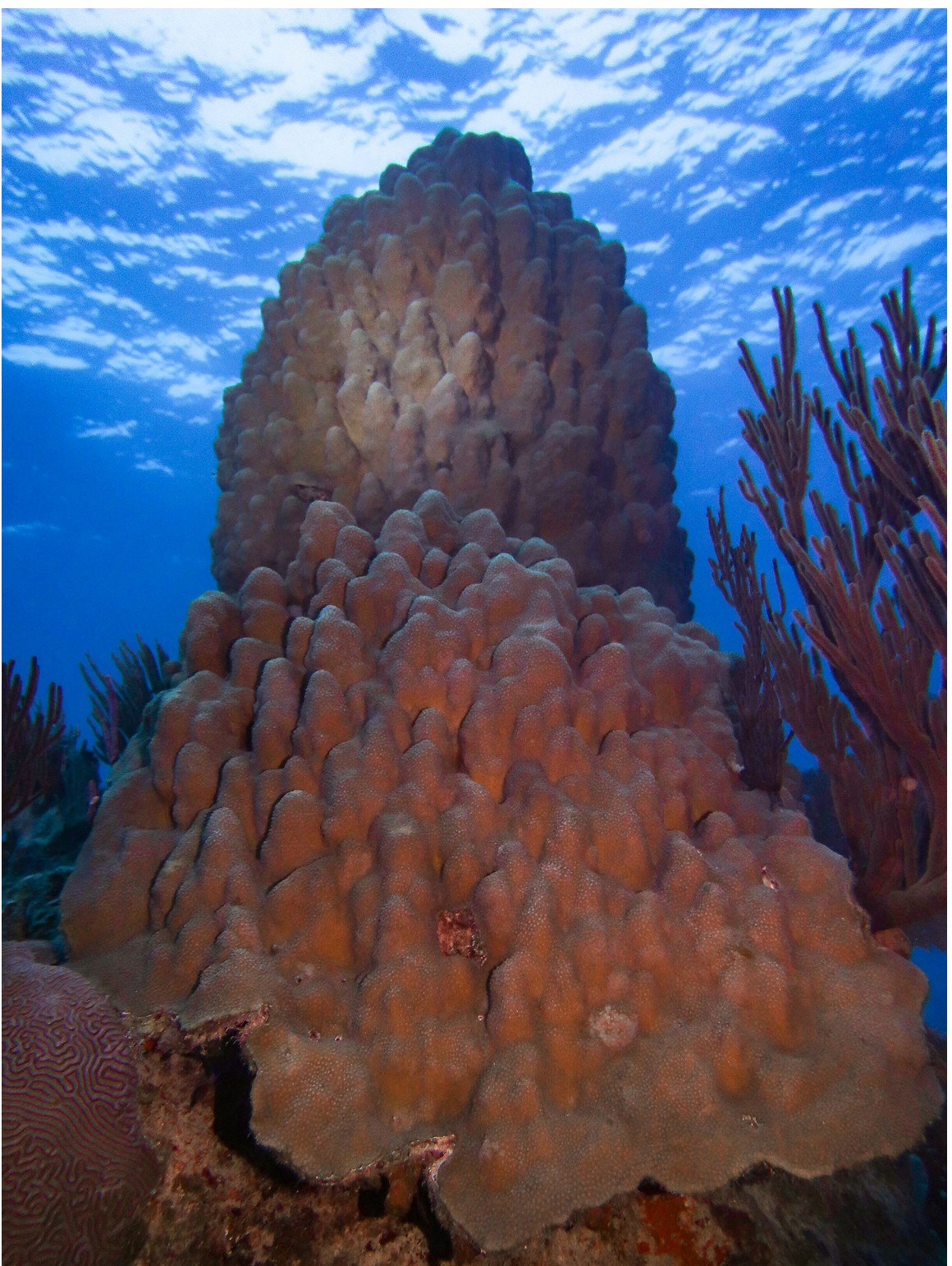

**Fig 2. *Orbicella faveolata* in Cayo Arenas, Campeche Bank, Mexico.** ©Lorenzo Álvarez-Filip.

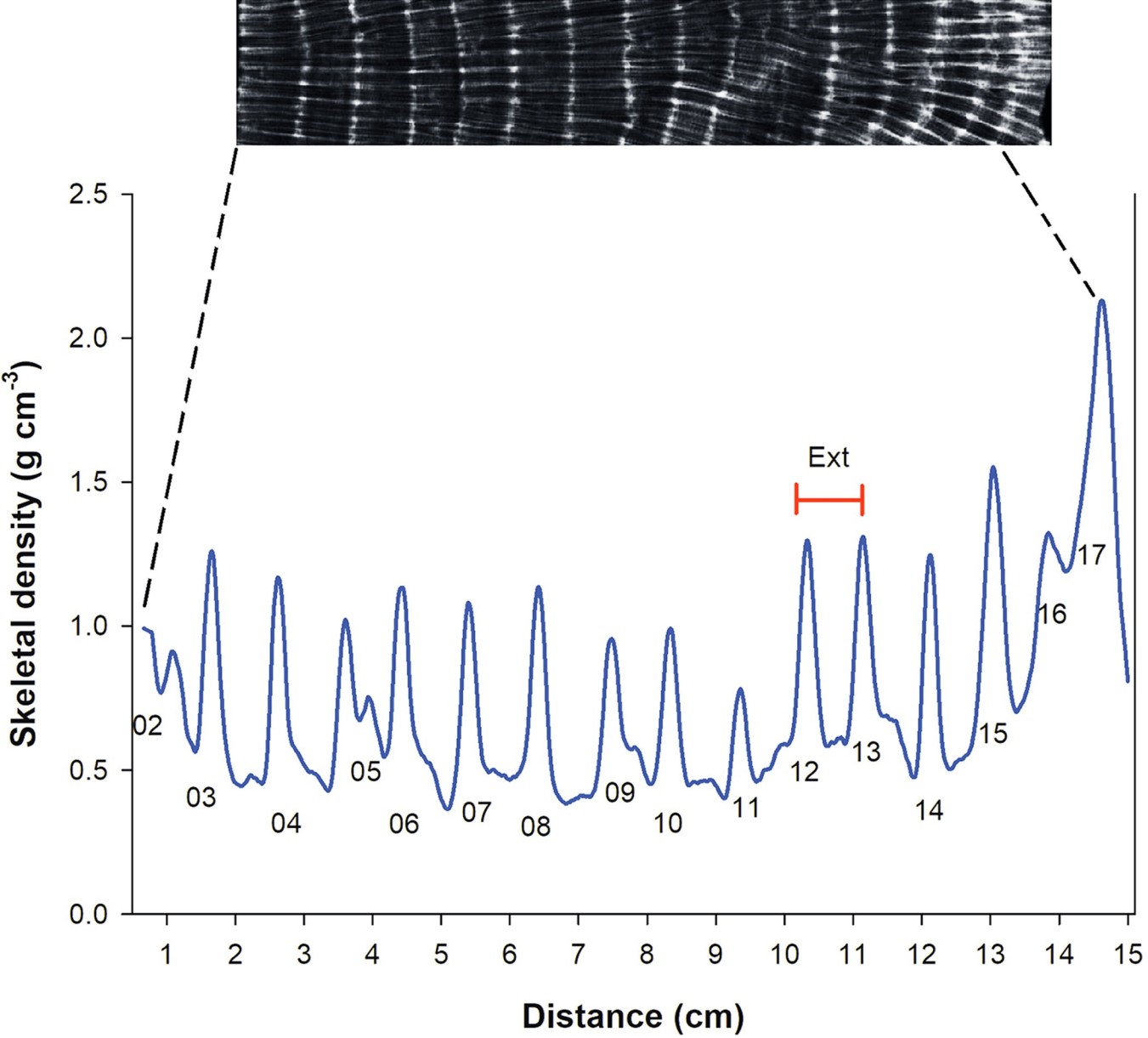

**Fig 3. Typical densitometry out put from digitized images of X-radiographs.** Skeletal density bands have been dated from the known date of collection. Measurements and calculations procedure for density, extension rate and calcification rate is further described in the text. Ext = extension rate.

[41]. Annual minimal/maximal, and mean SST values were calculated from monthly data for each year and were used to identify the relationship with annual coral growth rates over a 24-year period.

## Statistical analysis

For all sclerochronological characteristics, annual means, standard deviation (± SD), and range values were calculated by pooling data from all five coral colonies. To determine differences between years for each sclerochronological characteristic, we used Analysis of Variance (one-way ANOVA) and/or Kruskal-Wallis (ANOVA based on ranks). The relationship

between the sclerochronological characteristics (skeletal density, extension rate, and calcification rate) was evaluated using simple linear regression model ($R^2$). To assess the effect of temperature on *O. faveolata* growth a linear regression of the annual calcification rate as a function of annual minimum/maximum, and mean of SST was performed. Master index values for each parameter (density, extension, and calcification) were calculated by averaging the data from each yearly period pooled from all five colonies, followed by the application of a 3-year moving average according to Tomascik descriptions [42]. To identify the temporal patterns of the sclerochronological characteristics of *O. faveolata*, we analyzed the master chronological indexes and the maximum SST over the period of 1992–2016 using linear regression models. All statistical analyses were performed using a 95% confidence interval (α = 0.05), and all statistical analyses were performed using R studio (R Core Team 2018).

The permit for sample collection was provided by Secretaría de Agricultura, Ganadería, Desarrollo Rural, Pesca y Alimentación (SAGARPA, Permiso de Pesca de Fomento No. PPF/DGOPA-042/13).

## Results

Coral growth characteristics of *O. faveolata* were obtained from 153 pairs of density bands over a 25-year interval (1992–2016); showing a mean skeletal density, extension rate and calcification rate of $1.04 \pm 0.30$ g cm$^{-3}$, $0.82 \pm 0.08$ cm yr$^{-1}$ and $0.85 \pm 0.18$ g cm$^{-2}$ yr$^{-1}$, respectively. During the time studied, the skeletal density showed a significant difference ($F = 2.665$, $P<0.001$); 2001 showed the lowest value in skeletal density with 0.70 cm$^{-3}$ in contrast with 1995, which exhibited the highest with 1.87 cm$^{-3}$. While the extension rate did not show any significant difference ($F = 0.331$, $P<0.998$), however, the lowest value was during 1995 (0.59 cm year$^{-1}$) and the highest in 2002 (0.95 cm year$^{-1}$). Also, the calcification rate did not show any significant difference ($H = 28.042$, $P = 0.258$), exhibiting the lowest value in 2001 and the highest in 1995 (Table 1; S1 Appendix).

Relationships between sclerochronological parameters were significantly positive for skeletal density and calcification rate ($R^2 = 0.819$, $p<0.001$; Fig 4A). In contrast, the extension rate and the skeletal density resulted in a slight negative relation ($R^2 = 0.338$, $p = 0.002$; Fig 4C). The only significant relationship between temperature and calcification rate was with the maximal SST, showing a negative relationship ($R^2 = 0.173$, $p = 0.038$), as SST increases, the *O. faveolata* calcification rate tends to decrease over time (Fig 5B). While with the mean and minimal SST, no relationship was found with the *O. faveolata* calcification rate, Fig 5A and 5C.

The master chronological index allowed us to observe the trend of the three sclerochronological variables and the maximum SST throughout the years. The master indices of calcification rate and skeletal density in *O. faveolata* show a tendency to significant decrease since 1994 (Fig 6B and 6C), while the master index of extension rate increased in 1996 (Fig 6A). From 1994 to 2016 *O. faveolata* has had a 33% reduction in its calcification rate, but the highest reduction was during 1994 to 2001 with 59%. While the master index of the maximum SST tended to increase, showing values above its mean during 1996–1999 and 2008–2016 (Fig 6D).

## Discussion

In the last four decades, coral reefs worldwide have been increasingly affected by extreme temperatures, leading to different trajectories of coral bleaching, mortality, and a decrease in the calcification rate of corals [30, 43, 44]. This is representing the first report of the sclerochronological characteristics of *O. faveolata* from Cayo Arenas and the effect of seawater temperature over the past 25 years. The result here provides evidence for the impact of increased SST on calcification rates of *O. faveolata* at an isolated reef location in the Gulf of Mexico.

**Table 1. Mean annual and their standard deviation of skeletal density (g cm⁻³), extension rate (cm year⁻¹) and calcification rate (g cm⁻² year⁻¹) of *Orbicella faveolata*, mean annual of maximum SST and thermal anomalies (˚C) in Cayo Arenas.** During 1992–2016 in Cayo Arenas.

| Years | Density | Extension rate | Calcification rate | Maximum SST | Thermal anomalies |
|---|---|---|---|---|---|
| 2016(15) | 0.94 ± 0.22 | 0.88 ± 0.36 | 0.80 ± 0.34 | 29.97 | 0.39 |
| 2015(15) | 0.80 ± 0.24 | 0.86 ± 0.21 | 0.64 ± 0.08 | 29.99 | 0.41 |
| 2014(15) | 0.76 ± 0.16 | 0.90 ± 0.32 | 0.66 ± 0.22 | 29.75 | 0.17 |
| 2013(15) | 0.77 ± 0.15 | 0.86 ± 0.25 | 0.66 ± 0.23 | 29.65 | 0.07 |
| 2012(15) | 0.81 ± 0.20 | 0.76 ± 0.16 | 0.60 ± 0.18 | 29.49 | -0.09 |
| 2011(15) | 0.85 ± 0.34 | 0.85 ± 0.22 | 0.69 ± 0.19 | 29.68 | 0.10 |
| 2010(15) | 0.91 ± 0.38 | 0.85 ± 0.29 | 0.71 ± 0.22 | 29.79 | 0.21 |
| 2009(15) | 0.78 ± 0.16 | 0.85 ± 0.17 | 0.64 ± 0.17 | 29.71 | 0.13 |
| 2008(15) | 0.80 ± 0.20 | 0.85 ± 0.20 | 0.68 ± 0.31 | 29.48 | -0.10 |
| 2007(15) | 0.94 ± 0.43 | 0.94 ± 0.28 | 0.80 ± 0.26 | 29.36 | -0.22 |
| 2006(15) | 0.91 ± 0.30 | 0.77 ± 0.26 | 0.70 ± 0.35 | 29.50 | -0.08 |
| 2005(15) | 0.88 ± 0.30 | 0.91 ± 0.28 | 0.82 ± 0.47 | 29.59 | 0.01 |
| 2004(15) | 0.90 ± 0.27 | 0.86 ± 0.33 | 0.72 ± 0.30 | 29.55 | -0.03 |
| 2003(15) | 0.94 ± 0.32 | 0.82 ± 0.28 | 0.75 ± 0.34 | 29.49 | -0.09 |
| 2002(15) | 1.05 ± 0.38 | 0.95 ± 0.35 | 1.01 ± 0.60 | 29.53 | -0.05 |
| 2001(12) | 0.70 ± 0.23 | 0.68 ± 0.12 | 0.52 ± 0.08 | 29.50 | -0.08 |
| 2000(09) | 0.94 ± 0.40 | 0.85 ± 0.17 | 0.70 ± 0.22 | 29.52 | -0.06 |
| 1999(09) | 1.04 ± 0.51 | 0.87 ± 0.41 | 0.76 ± 0.23 | 29.62 | 0.04 |
| 1998(09) | 1.24 ± 0.53 | 0.85 ± 0.32 | 0.91 ± 0.33 | 29.80 | 0.22 |
| 1997(09) | 1.38 ± 0.54 | 0.75 ± 0.21 | 0.98 ± 0.17 | 29.71 | 0.13 |
| 1996(09) | 1.29 ± 0.34 | 0.84 ± 0.20 | 1.06 ± 0.24 | 29.55 | -0.03 |
| 1995(06) | 1.87 ± 0.20 | 0.59 ± 0.12 | 1.06 ± 0.06 | 29.21 | -0.37 |
| 1994(06) | 1.59 ± 0.07 | 0.76 ± 0.22 | 1.19 ± 0.27 | 29.34 | -0.24 |
| 1993(06) | 1.47 ± 0.09 | 0.72 ± 0.21 | 1.04 ± 0.34 | 29.30 | -0.28 |
| 1992(06) | 1.49 ± 0.08 | 0.73 ± 0.05 | 1.06 ± 0.05 | 29.45 | -0.13 |
| **Mean** | 1.04 ± 0.30 | 0.82 ± 0.08 | 0.85 ± 0.18 | 29.58 | - |

Thermal anomaly is considered temperatures +0.5 above the maximum average. In parenthesis the number of density bands used for each year.

Highlighting, a negative relationship between the coral calcification rate and the maximal SST of Cayo Arenas, in addition to the fact that the years that show positive thermal anomalies are the same ones that show a decrease in the rate of calcification, which coincides with the reductions in this sclerochronological parameter (20%) reported by Carricart-Ganivet et al. [8] when thermal conditions are suboptimal.

The annual values obtained for the three sclerochronological characteristics of *O. faveolata* are within the range of values compared with other reef areas in the tropical Atlantic region [23, 24, 45, 46]. The growth of scleractinian corals depends on the intrinsic characteristics of the species and the exogenous factors that influence their growth, they can invest calcification resources in building denser or more extensive skeletons [25, 40]. In the corals of the *Orbicella* genus, their growth strategy is to invest calcification resources in building denser skeletons [25, 47]. In this work, the relationship of the sclerochronological characteristics shows that *Orbicella faveolata* uses its calcification resources to make denser skeletons. This strategy of having denser skeletons makes the colonies less susceptible to fragmentation but more susceptible to bioerosion since some organisms prefer more resistant microhabitats that allow them to protect themselves against their predators [25, 48, 49]. On the other hand, in this work, it is corroborated that the main causes of the decrease in *O. faveolata* calcification rate is because of

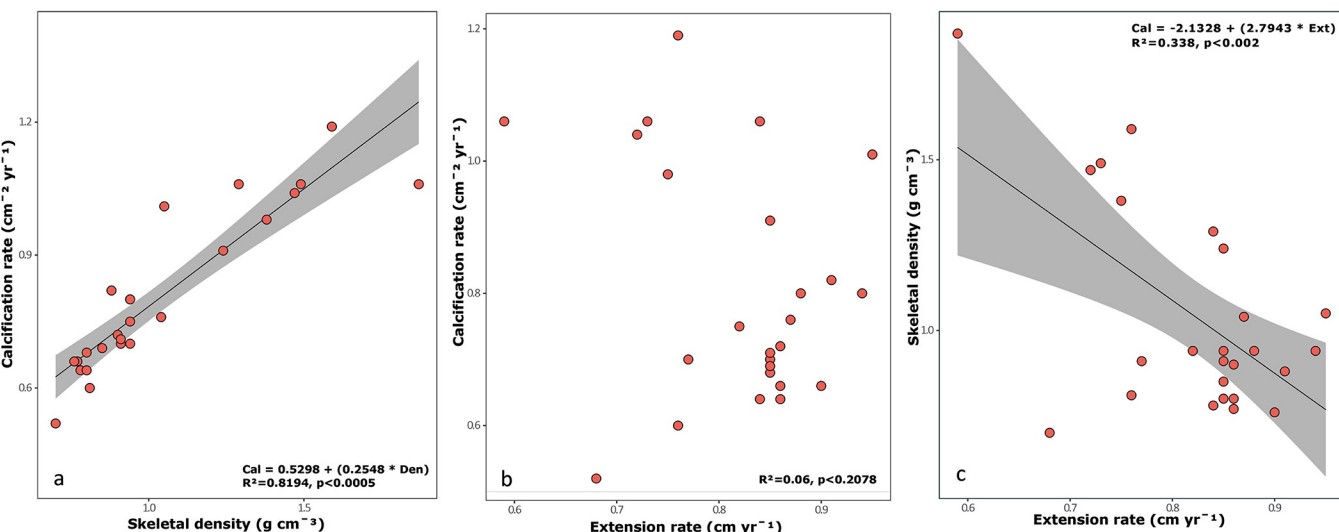

**Fig 4. Scatterplots of the three growth variables in *Orbicella faveolata*.** (a) relationship between calcification rate and skeletal density, (b) between calcification rate and extension rate, and (c) between extension rate and skeletal density. A regression line and equation are shown for the former relationship.

thermal anomalies derived from climate change, even removing local stressors, the increase in SST may disrupt important physiological processes, such as growth and coral calcification [31, 50–52]. Although the colonies of *O. faveolata* studied in Cayo Arenas are far from direct anthropic impact (pollution, coastal development, tourism), the index master shows that in the last 20 years, *O. faveolata* has had also significant changes in calcification rate (decreasing 33%) associated with thermal anomalies, occurring most frequent and intense from 2000 to 2016. This decrease in the rate of calcification can be considered a sign that the corals are under stress, even in conditions where the corals have a high cover and in which high mortality has yet not occurred as in other reefs [53, 54].

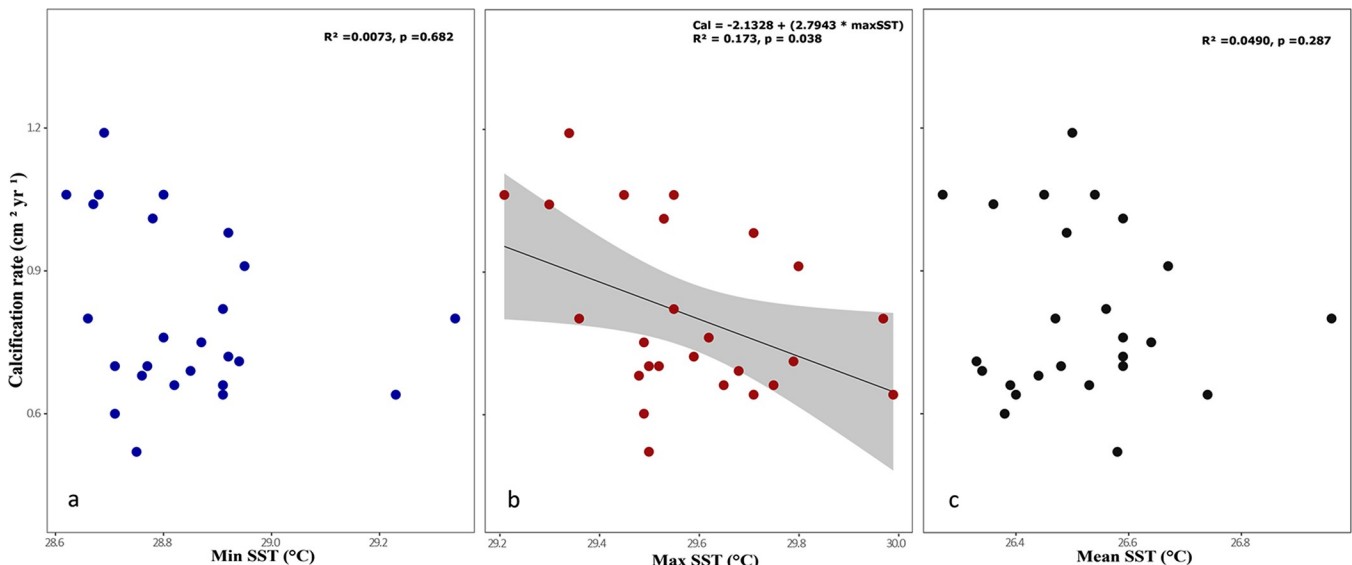

**Fig 5. The relationship between sea surface temperature (SST) and calcification rate of *Orbicella faveolata* in Cayo Arenas, Campeche Bank.** Relationship between calcification rate with (a) minimum SST, (b) maximum SST, and (c) mean SST. A regression line and equation are shown for the former relationship.

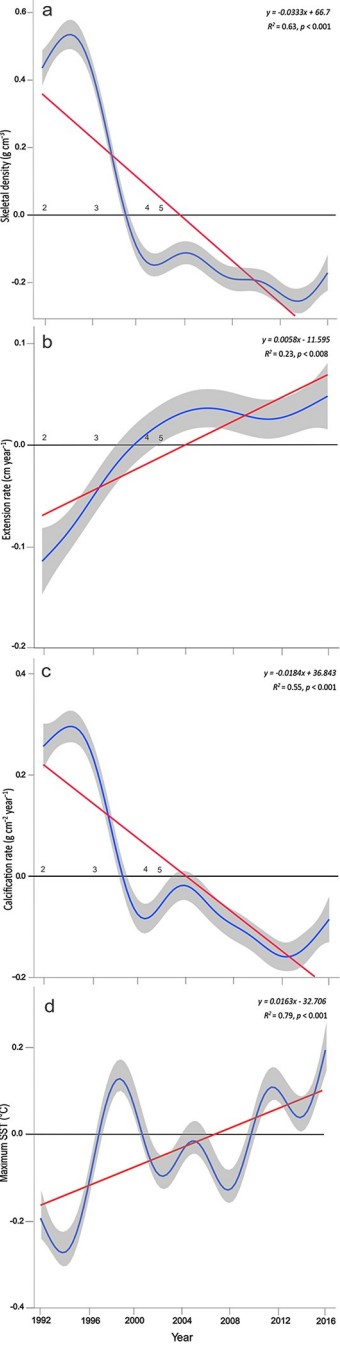

**Fig 6. Master chronology index (n = 5 corals) of *O. faveolata* from 1992–2016.** Annual coral growth characteristics exhibit significant trends (red lines) over the 24-year period: (a) extension rate, (b) skeletal density, (c) calcification rate, and (d) maximum sea surface temperature. A regression line and equation are shown for the former relationship. The blue lines represent the mean annual values (± 95% confidence interval, shown as shaded lines) of each sclerochronological variable using all years. The numbers along the horizontal axis indicate the number of colonies used for each year.

The index master chronologies provide details on the trends in growth parameters of *O. faveolata* since 1999, skeletal density and calcification rates show values below the reference annual averages, but skeletal extension values have remained above the average from 1997 onwards. This increase in the skeletal extension rate is known as "stretching modulation of skeletal growth" [21] (Carricart-Ganivet and Merino, 2001), which has been reported in other studies, where *O. faveolata* responds to thermal and environmental stress, extending their skeletons and decreasing their skeletal density and calcification rate [21, 24, 46, 47]. The shift in energy resources for *O. faveolata* implies that instead of utilizing its calcification resources to create denser skeletons, it utilizes them to create more extensive skeletons. This change could potentially result in colonies with structurally fragile skeletons, subsequently reducing their accretion rate and overall contribution to the physical maintenance of the coral reef ecosystem [55, 56]. This strategy is specific to the genus and the species, however, over time this strategy can be modified by thermal anomalies, for example, when a year with greater extension and lower density occurs, it may be the biological response in responding to an anomaly [24, 46].

*Orbicella* species are one of the main coral reef-building species in the tropical Atlantic and have a significant accretion threshold [57]. Their calcification plays an important role in maintaining and balancing the physical and biological structure of the reef [58, 59], resulting in the production of large amounts of $CaCO_3$. However, our findings reveal a negative relationship between calcification rate and maximum SST, with a 33% reduction in calcification rate during the period studied. This indicates that *O. faveolata* from Cayo Arenas is threatened by increased SST. Under the context of climate predictions, including global warming, anticipated rise in SST ($>2°C$), and more frequent thermal anomalies, the historical data presented herein could suggest that coral growth and calcification rates may exacerbate in the coming decades [60, 61]. The potential loss of reef-building coral species would have serious consequences for the future persistence of the physical framework and ecological functionality of coral reefs in the Mexican Atlantic region [20, 62].

## Supporting information

**S1 Appendix.**
(XLSX)

## Acknowledgments

We greatly thank to Esmeralda Pérez Cervantes and Nuria A. Saldivar Estrada for their logistical support during fieldwork.

## Author Contributions

**Conceptualization:** D. Wendoline Sánchez-Pelcastre, Israel Cruz-Ortega, Juan P. Carricart-Ganivet.

**Formal analysis:** D. Wendoline Sánchez-Pelcastre, J. J. Adolfo Tortolero-Langarica, Juan P. Carricart-Ganivet.

**Funding acquisition:** Lorenzo Alvarez-Filip, Juan P. Carricart-Ganivet.

**Supervision:** Juan P. Carricart-Ganivet.

**Writing – original draft:** D. Wendoline Sánchez-Pelcastre, J. J. Adolfo Tortolero-Langarica, Lorenzo Alvarez-Filip, Juan P. Carricart-Ganivet.

**Writing – review & editing:** J. J. Adolfo Tortolero-Langarica, Lorenzo Alvarez-Filip, Juan P. Carricart-Ganivet.

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
