## [Decision Letter · Decision Letter 0]

26 Jun 2023

PONE-D-23-05903Sclerochronological characteristics of Orbicella faveolata in Cayo Arenas, a remote coral reef from the Gulf of MexicoPLOS ONE

Dear Dr. Carricart-Ganivet,

Thank you for submitting your manuscript to PLOS ONE. After careful consideration, we feel that it has merit but does not fully meet PLOS ONE’s publication criteria as it currently stands. Therefore, we invite you to submit a revised version of the manuscript that addresses the points raised during the review process. Both reviewers have made good suggestions. In particular please note the following.Inclusion of a radiograph (s) as a figure representing your the three sclerochronological features as suggested by Reviewer #1 seems appropriate. Please review the data availability requirements of PLOS ONE. As noted by Reviewer #2, more data needs to be made available. This needs to be done in order for this paper to be published by PLOS ONE. Reviewer #2 pointed out that there are several problems with the statistical analyzes.  The English of the manuscript is very good, but not quite perfect. As reviewer #2 points out, this needs to be fixed. Please note that PLOS ONE doesn't provide language editing. Please submit your revised manuscript by Aug 10 2023 11:59PM. If you will need more time than this to complete your revisions, please reply to this message or contact the journal office at plosone@plos.org. Please include the following items when submitting your revised manuscript:A rebuttal letter that responds to each point raised by the academic editor and reviewer(s). You should upload this letter as a separate file labeled 'Response to Reviewers'.A marked-up copy of your manuscript that highlights changes made to the original version. You should upload this as a separate file labeled 'Revised Manuscript with Track Changes'.An unmarked version of your revised paper without tracked changes. You should upload this as a separate file labeled 'Manuscript'.

We look forward to receiving your revised manuscript.

Kind regards,

Erik V. Thuesen, Ph.D.

Academic Editor

PLOS ONE

“The present work was supported by the projects UNAM-DGAPA-PAPIIT IN200420 to JPCG, and UNAM-DGAPA-PAPIIT IG201323 to LAF. We greatly thank to Esmeralda Pérez Cervantes and Nuria A. Saldivar Estrada for their logistical support during fieldwork. The permit for sample collection was provided by Secretaría de Agricultura, Ganadería, Desarrollo Rural, Pesca y Alimentación (SAGARPA, Permiso de Pesca de Fomento No. PPF/DGOPA-042/13).”

“JPCG UNAM-DGAPA-PAPIIT IN200420

LAF UNAM-DGAPA-PAPIIT IG201323

5. We note that [Figure 1] in your submission contain [map/satellite] images which may be copyrighted. All PLOS content is published under the Creative Commons Attribution License (CC BY 4.0), which means that the manuscript, images, and Supporting Information files will be freely available online, and any third party is permitted to access, download, copy, distribute, and use these materials in any way, even commercially, with proper attribution. For these reasons, we cannot publish previously copyrighted maps or satellite images created using proprietary data, such as Google software (Google Maps, Street View, and Earth). For more information, see our copyright guidelines: http://journals.plos.org/plosone/s/licenses-and-copyright.

Natural Earth (public domain): http://www.naturalearthdata.com/.

Reviewers' comments:

Reviewer's Responses to Questions

**Comments to the Author**

1. Is the manuscript technically sound, and do the data support the conclusions?

Reviewer #1: Yes

Reviewer #2: Yes

2. Has the statistical analysis been performed appropriately and rigorously? 

Reviewer #1: Yes

Reviewer #2: No

3. Have the authors made all data underlying the findings in their manuscript fully available?

Reviewer #1: Yes

Reviewer #2: No

4. Is the manuscript presented in an intelligible fashion and written in standard English?

Reviewer #1: Yes

Reviewer #2: No

5. Review Comments to the Author

Reviewer #1: Authors has to mention the protection status of coral and its importance in the introduction section

Line 63 - Change the word to “in understanding/ to understand”.

Line 72 – 73 - Rephrase the sentence for better clarity.

Line 79 - Change the sentence for better understanding.

Figure 4 - Provide regression value for minimum and mean SST.

Figure 5 - Give caption for representation of blue lines in the figure.

Reviewer #2: In this manuscript the authors obtain growth skeletal parameters (linear extension, density and calcification ) from five different colonies of Orbicella spp. These cores were all collected from the remote reef of Cayo Arenas. Growth parameters were obtained from 2-dimensional x-rays following previously published methods. These growth data were coupled to annual SST (obtained through remote sensing) to evaluate how SST changes over time might be affecting growth of this massive reef-building coral. The authors found a significant negative linear relationship between maximum annual SST and coral calcification and how negative trends in skeletal growth have been observed from 1998 onwards.

Importantly, the periods of time where the growth change rate is larger seem to align with periods with abrupt positive thermal anomalies.

This study constitutes the first reported dataset from the reef-building Orbicella from Cayo Arenas, and its results are of importance to further understand how different corals respond to climate change and its thermal thresholds.

As such, I recommend this manuscript to be accepted after major revisions have been addressed as detailed below. In addition to these changes, I would recommend to thoroughly examine both the grammar and the narrative of the manuscript, as at times it feels hard to follow the authors on their science.

Major changes:

Abstract

Line 38: [there] will be [ sic] is a prediction and needs to be supported by evidence/references. This sentence does not make sense. Rewrite.

Introduction

Line 62: Add references for studies that look into stress bands and past climate events.

Line 65: References to this sentence (Sclerochronological records from corals are important to understand the effect of climate change on coral calcification) are not entirely adequate. There are other important papers than can be cited here and that have been omitted from the literature review. For example, Cantin et al. 2010 (DOI: 10.1126/science.1190182) or Cooper et al. 2012 (DOI: 10.1126/science.1214570) among many other that use coral growth records coupled to temperature data to explore past changes in calcification due to thermal stress. A more thorough review would be useful for the reader, not only with data from the Atlantic/Caribbean but also from other massive species from the Pacific. This can help contextualise how ocean warming affecting corals is a global effect, and this study adds evidence to attests this.

In the introduction, the second and third paragraphs need to be re-structured. In Line 59 (Second paragraph) there is a hint at the existence of “stress bands” as a result of coral sensitivity. This seems unconnected to the rest of the paragraph. Furthermore, in the third paragraph, from

Line 68 it focuses on thermal stress and how this leads to stress bands. Instead, it might be easier from the reader perspective to follow a structure similar to: i)Coral calcification is important to maintain net reef accretion. Calcification records can aid in reef management efforts (the first paragraph of the introduction as it is); ii) Coral growth is sensitive to several environmental factors, with temperature being the most important variable. Sensitivity of the coral symbiont to this variable produces changes in coral growth. This is the basis for coral sclerochronology; iii) Skeletal formation is characterised by annual couplets of high-density and low-density bands (the base of the second paragraph). iv) Add a paragraph with studies showing the relationship between temperature and coral growth. v) and iv) Focus on studies from Orbicella spp. and intro to location of study as it is in Lines74 to 96.

Material and Methods

Line137 and below: More information is needed on how SST was obtained and treated. SST obtained from HadISST. I think details on whether data was downloaded as daily, monthly or annual is needed. The study specifies that annual SST was obtained, but then annual minimum and maximum SSTs were calculated. Are these annual minimum and maximum SSTs actually the minimum/maximum SST from monthly temperatures or from daily temperatures over a given year? This is not clear to me. Also the area/pixel information over which the information was obtained (i.e., resolution of data, coordinates,...). This is fundamental for reproducible science and it is missing. Make this paragraph into a new section after “Coral sampling and analysis of sclerochronological variables” along the lines of “Environmental data acquisition”.

Line 140 and throughout the text: Change annual minimal and maximal to annual minimum/maximum SST. I would recommend to thoroughly revise the grammar of the manuscript.

Lines 150 and 151: I don’t understand what the authors mean by “Only thermal anomalies and maximum SST index master were determined for their relationship with calcification rate. What is maximum SST index master? Could you explain?

Lines 154 and 155: When referring to an “index master” of sclerochronological characteristics, do the authors mean a “master chronology”? The term index master is confusing. Maybe calling it “master index” instead would make more sense, grammatically.

Line 157: When describing the master dataset being sequentially averaged 3 years, do the authors mean they applied a moving average with a 3-year window? If so, please specify to address it with the correct statistical term.

Line 158: Start a new sentence after the “averaged 3 years”. In this new sentence you describe how you calculate growth anomalies from the master dataset. This is independent from the 3-year averaging and by being in a different sentence it is clearer for the reader.

Statistical analysis and results

Not all statistical analysis explored in the results section are included in the Methods and other seems to be not well clarified. Here, I follow the structure of the results sections and detail what needs to be addressed.

Lines 166 to172: Presumably differences across time on each growth parameter were calculated with One-way ANOVA/F-test and/or Kruskal-Wallis/H-test? This is nowhere specified on the statistical analysis in methods.

Lines 174 to 183: Relationships between sclerochronological parameters are reported as R2 yet in methods it is described as “The relationship between the sclerochronological characteristics (skeletal density, extension rate, and calcification rate) was evaluated using simple correlations” (Lines 145-147).

Simple correlations only measure the level of association/linearity between two variables (without assigning dependency to these variables) and it is reported as “r” usually. Some examples of correlations would be (Pearson’s, Spearman’s or Kendall’s). The authors fail to describe what simple correlation method they use for these variables.

In addition, the reporting of R2 suggest that they might have used a simple lineal regression instead.

My preference in this case would be to use a correlation method (rather than linear modelling), because this way it is not necessary to assign what growth variable is dependent and which one is independent. For example, is linear extension dependent on skeletal density?. However, relationships between growth variables in the literature have been reported following either statistical approach, and one is no inherently wrong. But authors need to be consistent and detail what test was performed and report accordingly.

Line 185: “Index master chronological” is confusing. Maybe changing by “Master chronological index” would aid in clarity throughout the manuscript.

Lines 188: The authors describe a significant decreasing trend in density of O. faveolata. What statistical test was performed here? It is not described in the statistics method sections. A linear regression of growth variables vs. time would suffice to justify the significancy of the trends described here.

Discussion

Line 197: Add more references to this first sentence for a more compelling and strong affirmation.

Line 203: Do the authors mean here positive thermal anomalies? As these can be negative too, but the data seems to support the effect of positive maximum SST anomalies over calcification change. Please, clarify.

Line 215: Denser skeletons make colonies more susceptible to bioerosion and Line 237-238: fragile skeletons (due to higher extension and reduced density) and more susceptible to bioerosion. These two sentences contradict themselves and it is unclear to me whether denser skeletons are more prone to bioerosion or the opposite.

Line 249-250: “Under the context of climate predictions, which indicate that

ocean temperatures continue to rise and threaten corals.” This sentence feels out of context. Unclear what is the point the authors are trying to make.

Line 252: “The models presented herein indicate that most coral will suffer marked deterioration in the coming decades”. This is a strong prediction that cannot be deducted from the results of this study. This study confirms that thermal stress might has caused reduction in skeletal calcification of O. faveolate in the last two decade, but conclusion to future response of these massive species in other locations, and, much less, in “most coral reefs” cannot be drawn. Please, rephrase.

Figures and Data

Figures 3 and 4: Adding SD (or SE as error bars to these datapoints will be informative for the reader and it is important to report.

Figure 5: Equally, adding a 95% confidence ribbon to each time-series will also provide important information. For example, seems like in 1998 growth variability is wider than in other periods (from Table 1). This in itself is a very interesting observation that can provide extra information and context for the readers.

Data availability: It would be a good addition to include single core growth parameters as SOM. As it stand, only the data from the master chronologies, on a 3-year averaged mean, is available with the manuscript. Providing all raw data (growth parameters from each coral and SSTs) is essential to ensure reproducibility efforts.

Other data: Equally, it might be interesting to include in the SOM all linear regression model performed with calcification, linear extension and density with SST, maximum SST and minimum SST. This is not necessary and just an addition. As such, feel free to ignore.

Minor changes:

Line 27: Local anthropic stressors. Anthropogenic might be a more common adjective to use here instead (and also throughout the manuscript)

Line 37: the maximum SST has been increased is confusing as an indirect sentence and suggest it has been increased by someone/something. . Use a direct sentence for clarity: SST has increased.

Line 42: From living organisms rather than on living organisms.

Lines 42 to 45: This is one long sentence. Break into two or more sentences for clarity.

Line 45: Measurements would be more appropriate than measures. Technically both can be used as noun, but the former is more common. This is just a recommendation, feel free to ignore.

Line 64: “coral calcification and survival rates”.

Line 69: have become one most important threat to coral

Line 82: AND wave exposure.

Line 106: with an intra-annual SST, from 29.1°C during June to September and colder waters 23.5°C from January to April [35] This sentence needs to be re-structured and clearer.

Line 131: Include a sentence briefly describing the methods for density calculations used in Reference 38 (i.e., density measured over a 4x2 area every XX mm, or similar) so reader can understand this methods without having to swing between publications.

6. PLOS authors have the option to publish the peer review history of their article (what does this mean?). If published, this will include your full peer review and any attached files.

Reviewer #1: No

Reviewer #2: No

---

## [Author Response · Author response to Decision Letter 0]

18 Sep 2023

Reviewer #1: Authors has to mention the protection status of coral and its importance in the introduction section.

Line 63 - Change the word to “in understanding/ to understand”.

Authors’ response: “in understanding” was changed to “understand”.

Line 72 – 73 - Rephrase the sentence for better clarity.

Authors’ response: We appreciate the reviewer's comment The following sentences (lines 59-60) have been revised to provide greater specificity and clarity for ease of understanding.

…” corals may lose the capacity to maintain vital physiological process and may even perish.”

Line 79 - Change the sentence for better understanding.

Authors’ response: We appreciate the reviewer's comment. For sake if clarity, the following sentences (lines 86-89) were changed to:

“Yet, most of these reports have been conducted in reef locations or periods influenced by both natural and local anthropic factors, which makes it difficult to identify the level of influence of each factor.”

Figure 4 - Provide regression value for minimum and mean SST.

Authors’ response: Statistical information was added for all regressions (figure 4).

Figure 5 - Give caption for representation of blue lines in the figure.

Authors’ response: Attending the reviewer´s suggestion, the figure caption was modified to:

Fig 5. Master chronology index (n=5 corals) of O. faveolata from 1992–2016. Annual coral growth characteristics exhibit significant trends (red lines) over the 24-year period: (a) extension rate, (b) skeletal density, (c) calcification rate, and (d) maximum sea surface temperature. A regression line and equation are shown for the former relationship. The blue lines represent the mean annual values (± 95% confidence interval, shown as shaded lines) of each sclerochronological variable using all years. The numbers along the horizontal axis indicate the number of colonies used for each year.

Reviewer #2: In this manuscript the authors obtain growth skeletal parameters (linear extension, density and calcification) from five different colonies of Orbicella spp. These cores were all collected from the remote reef of Cayo Arenas. Growth parameters were obtained from 2-dimensional x-rays following previously published methods. These growth data were coupled to annual SST (obtained through remote sensing) to evaluate how SST changes over time might be affecting growth of this massive reef-building coral. The authors found a significant negative linear relationship between maximum annual SST and coral calcification and how negative trends in skeletal growth have been observed from 1998 onwards. Importantly, the periods of time where the growth change rate is larger seem to align with periods with abrupt positive thermal anomalies.

This study constitutes the first reported dataset from the reef-building Orbicella from Cayo Arenas, and its results are of importance to further understand how different corals respond to climate change and its thermal thresholds. 

As such, I recommend this manuscript to be accepted after major revisions have been addressed as detailed below. In addition to these changes, I would recommend to thoroughly examine both the grammar and the narrative of the manuscript, as at times it feels hard to follow the authors on their science. 

Authors’ response to reviewer´s general comments: We would like to thank the reviewer for their valuable comments aimed at improving our manuscript. In general, we carefully considered all comments and recommendations, and we believe that we have appropriately addressed them in this revised version of the manuscript.

Major changes: 

Abstract

Comment #1. Line 38: [there] will be [sic] is a prediction and needs to be supported by evidence/references. This sentence does not make sense. Rewrite. 

Authors response: We are agreeing with the reviewer´s comment, for sake of clarity the sentence (lines 39-42) was modified to:

“If the temperature continues to rise, there is a conceivable risk of experiencing a decline in reef-building coral species. This scenario, in turn, could pose a significant threat, endangering not only the framework of coral reefs but also their ecological functionality, even within remote Atlantic reef ecosystems.”

Introduction

Comment #2. Line 62: Add references for studies that look into stress bands and past climate events. 

Authors’ response: We have incorporated the next reference into the new version of the manuscript (line 77). This addition complements and strengthens the fluency of the text and the context of the study.

DeCarlo TM, Cohen AL. Dissepiments, density bands and signatures of thermal stress in Porites skeletons. Coral Reefs. 2017; 36:749–761. Available from: https://doi.org/10.1007/s00338-017-1566-9

Comment #3. Line 65: References to this sentence (Sclerochronological records from corals are important to understand the effect of climate change on coral calcification) are not entirely adequate. There are other important papers than can be cited here and that have been omitted from the literature review. For example, Cantin et al. 2010 (DOI: 10.1126/science.1190182) or Cooper et al. 2012 (DOI: 10.1126/science.1214570) among many other that use coral growth records coupled to temperature data to explore past changes in calcification due to thermal stress. A more thorough review would be useful for the reader, not only with data from the Atlantic/Caribbean but also from other massive species from the Pacific. This can help contextualize how ocean warming affecting corals is a global effect, and this study adds evidence to attests this. 

Authors’ response: We appreciate the literature recommended by the reviewer. We have incorporated a new reference into the new version of the manuscript (line 60 and 62).

Cooper TF, O´leary RA, Lough JM. Growth of Western Australian Corals in the Anthropocene. 2012. Science335,593-596. https://doi.org/10.1126/science.1214570

Comment #4. In the introduction, the second and third paragraphs need to be re-structured. In Line 59 (Second paragraph) there is a hint at the existence of “stress bands” as a result of coral sensitivity. This seems unconnected to the rest of the paragraph. Furthermore, in the third paragraph, from Line 68 it focuses on thermal stress and how this leads to stress bands. Instead, it might be easier from the reader perspective to follow a structure similar to i) Coral calcification is important to maintain net reef accretion. Calcification records can aid in reef management efforts (the first paragraph of the introduction as it is); ii) Coral growth is sensitive to several environmental factors, with temperature being the most important variable. Sensitivity of the coral symbiont to this variable produces changes in coral growth. This is the basis for coral sclerochronology; iii) Skeletal formation is characterized by annual couplets of high-density and low-density bands (the base of the second paragraph). iv) Add a paragraph with studies showing the relationship between temperature and coral growth. v) and iv) Focus on studies from Orbicella spp. and intro to location of study as it is in Lines 74 to 96. 

Authors’ response: We appreciate the valuable comments aimed at improving the structure of our introduction. The introductory paragraphs have been modified in order to make the reading clearer and more concise (lines 44- 105 to read)

“Environmental reconstructions can be obtained on living organisms such as scleractinian corals, which may provide crucial information on the natural response and acclimatization/adaptation of species, leading to a better understanding and application of management and conservation measures in ecosystems to future climate scenarios [1, 2]. Growth in reef-building corals results from the accumulation of large amounts of calcium carbonate (CaCO3) in their skeletons. This is known as coral calcification and it is the leading process that builds and maintains both the physical framework and the balance ecological functionality of the reef ecosystems [3, 4].

Coral calcification is mediated by environmental factors such as light irradiation, water temperature, water chemistry, nutrient concentration, and others [5]. However, the seawater temperature is one of the most important variables that control the variation in calcification rates and skeletal growth of scleractinian corals, and it may determine their distribution along spatial gradients [5 – 8]. However, prolonged exposure to a temperature that is +1°C above the species threshold can cause thermal stress to coral organisms, leading to the expulsion of the endosymbiotic algae (Symbiodinium) in an event known as coral bleaching. If the stress persists for more than four weeks, corals experience a diminished capacity for maintain vital physiological process and may even perish [9 – 11]. In the context of rapidly changing climatic conditions, temperature increases have become one of the most important threats to coral growth, calcification, and survival [9]. 

During the process of coral calcification and skeletal growth, corals form high- and low-density bands of calcium carbonate (g cm-3) between the summer and winter seasons, respectively [11 – 13]. These bands act as natural "sclerochronometers" similar to tree rings in dendrochronology [12, 14], allowing the history of tropical ocean and coral reef ecosystems to be traced through the calcium carbonate skeletons of reef-building corals [11, 15]. By analyzing the annual rhythmic patterns in the coral skeleton, we can reconstruct the coral's life history and understand how these organisms respond to environmental factors through coral growth characteristics such as skeletal density (g cm-3), extension rate (cm yr-1), and calcification rate (g cm-2 yr-1) rate [11, 15, 16]. Sclerocronology is an important tool in understanding the effects of climate change on coral calcification rates and the historical response of coral reef species [11, 12 ,17, 18]. Furthermore, the study of coral growth can be used to understand past climate events, such as thermal anomalies, pH, and nutrient concentration, through the analysis of atypical high-density bands known as "stress bands," which result from the sensitivity response of corals species to environmental variability [17].

Over the past twenty years, many studies have indicated a reduction of 11-21% in calcification rates within the tropical regions of the Great Barrier Reef located in Australia [9] Nonetheless, considering their life cycles and adaptation mechanisms, individual species exhibit distinct growth rates influenced by local and regional environmental factors [5,18, 19]. It is plausible that corals might adapt differently across many reef regions. In the west-Atlantic and Caribbean region, the massive reef-building coral Orbicella spp. significantly contribute to the formation and maintenance of coral reef ecosystems [20]. Because of its morphological characteristics and its abundance, this species is one of the coral species most used for sclerochronological reconstruction in the region. Yet, most of these reports have been conducted in reef locations or periods influenced by both natural and local anthropic factors, which difficult to identify the source of variability [7, 8, 21, 22, 23, 24]. The local anthropogenic effects are important, as in addition to temperature, coral growth is influenced by the variability of environmental factors such as nutrient load [25, 26], sedimentation [21], depth [27], wave exposure [28]. Therefore, estimating calcification and growth rates in remote locations isolated from local human influence would allow us to clearly see the effects of climate change. 

Cayo Arenas in the Campeche Bank, Gulf of Mexico is a small reef bank situated ~170 km from the coast north of the Yucatan Peninsula. This reef can be considered a “natural laboratory" due to its remoteness, this site presents the possibility of studying the natural dynamics of biological and ecological aspects far from local anthropogenic pressures, as well as assessing the direct effects of climate change on coral reefs. The objective of this study was to establish long-term baselines of growth (skeletal density, extension rate, and calcification rate) of O. faveolata from Cayo Arenas, by evaluating historical coral growth trends from 1992 to 2016, and the effect of seawater temperature on calcification rates. These results underline the need to improve our understanding of the role that temperature plays in the growth of scleractinian corals in the Atlantic region, considering that the increase in ocean temperature and global climate changes may exacerbate coral reef degradation [29 – 31].”

Material and Methods

Comment #5. Line137 and below: More information is needed on how SST was obtained and treated. SST obtained from HadISST. I think details on whether data was downloaded as daily, monthly or annual is needed. The study specifies that annual SST was obtained, but then annual minimum and maximum SSTs were calculated. Are these annual minimum and maximum SSTs actually the minimum/maximum SST from monthly temperatures or from daily temperatures over a given year? This is not clear to me. Also, the area/pixel information over which the information was obtained (i.e., resolution of data, coordinates...). This is fundamental for reproducible science, and it is missing. Make this paragraph into a new section after “Coral sampling and analysis of sclerochronological variables” along the lines of “Environmental data acquisition”. 

Authors’ response: We appreciate the reviewer's recommendation. The following sentences (lines 147-154) have been revised to provide greater specificity and clarity for ease of understanding.

“Environmental data acquisition

The sea surface temperature (SST) was used as an environmental metric. Monthly SST dataset from 1992 to 2016 of Campeche Bank was obtained from satellite images (1° latitude-longitude grid resolution) acquired from the Hadley Centre Sea Ice and SST (HadISST, https://www.metoffice.gov.uk/hadobs/), dataset produced by the United Kingdom Meteorological Office [41]. Annual minimum/maximum, and mean SST values were calculated from monthly data for each year and were used to identify the relationship with annual coral growth rates over a 24-year period. To determine the thermal anomalies of the maximum SST, the data of the maximum SST from the study period (1992-2016) were integrated and then, the total mean of the maximum SST of each year was subtracted.”

Comment #6. Line 140 and throughout the text: Change annual minimal and maximal to annual minimum/maximum SST. I would recommend to thoroughly revise the grammar of the manuscript. 

Authors’ response: We are agreeing with the reviewer´s recommendation, minimum/maximum SST is used throughout the new manuscript text.

Comment #7. Lines 150 and 151: I don’t understand what the authors mean by “Only thermal anomalies and maximum SST index master were determined for their relationship with calcification rate. What is maximum SST index master? Could you explain?

Authors’ response: We apologize for these lines; they shouldn't be there due to an inadvertent mistake.

Comment #8. Lines 154 and 155: When referring to an “index master” of sclerochronological characteristics, do the authors mean a “master chronology”? The term index master is confusing. Maybe calling it “master index” instead would make more sense, grammatically. 

Authors’ response: Yes, indeed, we are referring to the master chronology. To enhance clarity, we have replaced 'index master' with 'master chronological index' throughout the manuscript.

Comment #9. Line 157: When describing the master dataset being sequentially averaged 3 years, do the authors mean they applied a moving average with a 3-year window? If so, please specify to address it with the correct statistical term. 

Authors’ response: We appreciate the reviewer´s observation. To avoid misperception, the sentence (lines 165-168) was modified to:

“Master index values for each parameter (density, extension, and calcification) were calculated by averaging the data from each yearly period pooled from all five colonies, followed by the application of a 3-year moving average according to Tomascik descriptions [42].”

Comment #10. Line 158: Start a new sentence after the “averaged 3 years”. In this new sentence you describe how you calculate growth anomalies from the master dataset. This is independent from the 3-year averaging and by being in a different sentence it is clearer for the reader. 

Authors’ response: We agree with the reviewer's comment. Since this sentence stands independently, we have moved it to the section on environmental data acquisition. Lines 148-154 to read:

” The sea surface temperature (SST) was used as an environmental metric. Monthly SST dataset from 1992 to 2016 of Campeche Bank was obtained from satellite images (1° latitude-longitude grid resolution) acquired from the Hadley Centre Sea Ice and SST (HadISST, https://www.metoffice.gov.uk/hadobs/), dataset produced by the United Kingdom Meteorological Office [41]. Annual minimum/maximum, and mean SST values were calculated from monthly data for each year and were used to identify the relationship with annual coral growth rates over a 24-year period. To determine the thermal anomalies of the maximum SST, the data of the maximum SST from the study period (1992-2016) were integrated and then, the total mean of the maximum SST of each year was subtracted”

Statistical analysis and results

Comment #11. Not all statistical analysis explored in the results section are included in the Methods and other seems to be not well clarified. Here, I follow the structure of the results sections and detail what needs to be addressed. 

Lines 157 to160: Presumably differences across time on each growth parameter were calculated with One-way ANOVA/F-test and/or Kruskal-Wallis/H-test? This is nowhere specified on the statistical analysis in methods. 

Authors response: We appreciate the reviewer's observation. We have added a description of the statistical analysis to the methods section. Lines 162-165 to read:

“For all sclerochronological characteristics, annual means, standard deviation (± SD), and range values were calculated by pooling data from all five coral colonies. To determine differences between years for each sclerochronological characteristic, we used Analysis of Variance (one-way ANOVA) and/or Kruskal-Wallis (ANOVA based on ranks)…”

Lines 174 to 183: Relationships between sclerochronological parameters are reported as R2 yet in methods it is described as “The relationship between the sclerochronological characteristics (skeletal density, extension rate, and calcification rate) was evaluated using simple correlations” (Lines 145-147). Simple correlations only measure the level of association/linearity between two variables (without assigning dependency to these variables) and it is reported as “r” usually. Some examples of correlations would be (Pearson’s, Spearman’s or Kendall’s). The authors fail to describe what simple correlation method they use for these variables. In addition, the reporting of R2 suggest that they might have used a simple lineal regression instead. My preference in this case would be to use a correlation method (rather than linear modelling), because this way it is not necessary to assign what growth variable is dependent and which one is independent. For example, is linear extension dependent on skeletal density? However, relationships between growth variables in the literature have been reported following either statistical approach, and one is no inherently wrong. But authors need to be consistent and detail what test was performed and report accordingly. 

Authors’ response: In this section, the necessary corrections have been applied to the description of the statistical analysis and the results. The coefficient of determination (r2) was considered to describe the relationships between growth parameters. Lines 160-165 to read:

“The relationship between the sclerochronological characteristics (skeletal density, extension rate, and calcification rate) was evaluated using simple linear regression model (R2). To assess the effect of temperature on O. faveolata growth a linear regression of the annual calcification rate as a function of annual minimum/maximum, and mean of SST was performed.”

Line 185: “Index master chronological” is confusing. Maybe changing by “Master chronological index” would aid in clarity throughout the manuscript. 

Authors’ response: We appreciate the reviewer's recommendation. The term 'Master Chronological Index' is used throughout the manuscript.

Lines 188: The authors describe a significant decreasing trend in density of O. faveolata. What statistical test was performed here? It is not described in the statistics method sections. A linear regression of growth variables vs. time would suffice to justify the significancy of the trends described here.

Authors’ response: We appreciate the reviewer's comment. Growth rate trajectories are depicted using linear regressions, this is a common approach used in skeletochronology studies. The significance of these trends, along with the inter-year variations, offers insights into the underlying causes of coral growth patterns. We have also included the details of the statistical test employed for analyzing growth tendencies (lines 168-170):

“To identify the temporal patterns of the sclerochronological characteristics of O. faveolata, we analyzed the master chronological indexes and the maximum SST over the period of 1992-2016 using linear regression models,”

Discussion 

Comment #12. Line 197: Add more references to this first sentence for a more compelling and strong affirmation. 

Authors’ response: We attend the reviewer´s recommendation. New references were added into the MS text, Line 205:

Carballo-Bolaños R, Soto D, Chen CA. Thermal Stress and Resilience of Corals in a Climate-Changing World. J. Mar. Sci. Eng. 2020; 8(15). Available from: https://doi.org/10.3390/jmse8010015.

Baker AC, Glynn PW, Riegl B. Climate change and coral reef bleaching: An ecological assessment of long-term impacts, recovery trends and future outlook. Estuarine, Coastal and Shelf Science. 2008; 80(4): 435-471. Available from: https://doi.org/10.1016/j.ecss.2008.09.003.

Line 203: Do the authors mean here positive thermal anomalies? As these can be negative too, but the data seems to support the effect of positive maximum SST anomalies over calcification change. Please, clarify. 

Authors’ response: Yes, it refers to the time when thermal anomalies are positive (maximum SST). For the sake of clarity, line 211 was modified to

“…in addition to the fact that the years that show positive thermal anomalies…”

Comment #13. Line 215: Denser skeletons make colonies more susceptible to bioerosion and Line 237-238: fragile skeletons (due to higher extension and reduced density) and more susceptible to bioerosion. These two sentences contradict themselves and it is unclear to me whether denser skeletons are more prone to bioerosion or the opposite. 

Authors response: We appreciate the reviewer's observation. To ensure clarity and avoid any incongruity, we have made the following modification to the sentence (lines 243-247):

“The shift in energy resources for O. faveolata implies that instead of utilizing its calcification resources to create denser skeletons, it utilizes them to create more extensive skeletons. This change could potentially result in colonies with structurally fragile skeletons, subsequently reducing their accretion rate and overall contribution to the physical maintenance of the coral reef ecosystem [55, 56].

Comment #14. Line 249-250: “Under the context of climate predictions, which indicate that ocean temperatures continue to rise and threaten corals.” This sentence feels out of context. Unclear what is the point the authors are trying to make. 

Authors’ response: We thank the reviewer´s comment. For sake of clarity the sentence (line 257-263) was changed to:

“Under the context of climate predictions, including global warming, anticipated rise in SST (>2°C), and more frequent thermal anomalies, the historical data presented herein could suggest that coral growth and calcification rates may exacerbate in the coming decades [60, 61].”

Comment #15. Line 252: “The models presented herein indicate that most coral will suffer marked deterioration in the coming decades”. This is a strong prediction that cannot be deducted from the results of this study. This study confirms that thermal stress might has caused reduction in skeletal calcification of O. faveolata in the last two decade, but conclusion to future response of these massive species in other locations, and, much less, in “most coral reefs” cannot be drawn. Please, rephrase. 

Authors’ response: To prevent misinterpretation and provide greater specificity, the conclusion paragraph (lines 256-268) was changed to:

“Orbicella species are one of the main coral reef-building species in the tropical Atlantic and have a significant accretion threshold [57]. Their calcification plays an important role in maintaining and balancing the physical and biological structure of the reef [58, 59], resulting in the production of large amounts of CaCO3. However, our findings reveal a negative relationship between calcification rate and maximum SST, with a 33% reduction in calcification rate during the period studied. This indicates that O. faveolata from Cayo Arenas is threatened by increased SST. Under the context of climate predictions, including global warming, anticipated rise in SST (>2°C), and more frequent thermal anomalies, the historical data presented herein could suggest that coral growth and calcification rates may exacerbate in the coming decades [60, 61]. The potential loss of reef-building coral species would have serious consequences for the future persistence of the physical framework and ecological functionality of coral reefs in the Mexican Atlantic region [62, 20].”

Figures and Data

Figures 3 and 4: Adding SD (or SE as error bars to these datapoints will be informative for the reader and it is important to report. 

Authors’ response: We appreciate the reviewer's comment; however, we believe that this might be redundant, as the standard deviation of annual growth parameters and SST is already reported for each year in Table 1. This data is the same data used for the relationship analysis in Figures 3 and 4.

Figure 5: Equally, adding a 95% confidence ribbon to each time-series will also provide important information. For example, seems like in 1998 growth variability is wider than in other periods (from Table 1). This in itself is a very interesting observation that can provide extra information and context for the readers. 

Authors’ response: We are agreeing with three reviewer suggestion. We added 95% interval confidence for each panel of figure 5:

Data availability: It would be a good addition to include single core growth parameters as SOM. As it stands, only the data from the master chronologies, on a 3-year averaged mean, is available with the manuscript. Providing all raw data (growth parameters from each coral and SSTs) is essential to ensure reproducibility efforts. 

Authors’ response: Annual growth parameters and sea surface temperatures data (±SD) for each year (from 1992 to 2016) are already provided in Table 1. Despite the data being pooled from all five colonies, this information is adequate for promoting further reproducibility and facilitating comparison efforts. However, the raw data will be made available to anyone who directly requests it from the authors.

Other data: Equally, it might be interesting to include in the SOM all linear regression model performed with calcification, linear extension and density with SST, maximum SST and minimum SST. This is not necessary and just an addition. As such, feel free to ignore. 

Authors’ response: Authors’ response: Statistical information was added in figures for all regressions.

---

## [Editor Report · Decision Letter 1]

11 Oct 2023

PONE-D-23-05903R1Sclerochronological characteristics of Orbicella faveolata in Cayo Arenas, a remote coral reef from the Gulf of MexicoPLOS ONE

Dear Dr. Carricart-Ganivet,

Thank you for submitting your revised manuscript to PLOS ONE. After careful consideration, we feel that it has merit but does not fully meet PLOS ONE’s publication criteria as it currently stands. Therefore, we invite you to submit a revised version of the manuscript that addresses the points raised during the review process.

In my previous message, I stated that the following were needed.

Inclusion of a radiograph (s) as a figure representing your the three sclerochronological features as suggested by Reviewer #1 seems appropriate. Please review the data availability requirements of PLOS ONE. As noted by Reviewer #2, more data needs to be made available. This needs to be done in order for this paper to be published by PLOS ONE.

I realize now that the suggestion about adding a radiograph was actually in the confidential comments to me, rather than a comment to you. I do not know why that would be confidential, but I missed that somehow. The suggestion was "provide radiographed images of the coral slabs and coral samples used in the study"; regardless, I think the inclusion of a radiograph would make the paper more accessible to scientists interested in corals that have not done these analyses themselves. I will leave that up to you, but if you have a good one handy, I think it will improve the paper.

Regarding the other comment of mine, you said in your reply to the reviewers that you could make that available upon request. PLOS ONE requires that you include that data either in a data repository or with the paper as an appendix. The journal doesn't subscribe to the "available upon request" data availability format.

Once you address these two issues, the manuscript should be fine for publication in PLOS ONE.

We look forward to receiving your revised manuscript.

Kind regards,

Erik V. Thuesen, Ph.D.

Academic Editor

PLOS ONE
---

## [Author Response · Author response to Decision Letter 1]

18 Oct 2023

Editor’s comments:

• Inclusion of a radiograph (s) as a figure representing your the three sclerochronological features as suggested by Reviewer #1 seems appropriate.

We added a figure (Fig. 3).

• Please review the data availability requirements of PLOS ONE. As noted by Reviewer #2, more data needs to be made available. This needs to be done in order for this paper to be published by PLOS ONE.

We added an Excel file of the full data as an appendix (Apendix 1) in the supplementary material.

---

## [Editor Report · Decision Letter 2]

20 Oct 2023

Sclerochronological characteristics of Orbicella faveolata in Cayo Arenas, a remote coral reef from the Gulf of Mexico

PONE-D-23-05903R2

Dear Dr. Carricart-Ganivet,

Thank you for sending the supporting information in the Excel file. We’re pleased to inform you that your manuscript has been judged scientifically suitable for publication and will be formally accepted for publication once it meets all outstanding technical requirements.

Kind regards,

Erik V. Thuesen, Ph.D.

Academic Editor

PLOS ONE
---

## [Editor Report · Acceptance letter]

31 Oct 2023

PONE-D-23-05903R2 

Sclerochronological characteristics of *Orbicella faveolata* in Cayo Arenas, a remote coral reef from the Gulf of Mexico 

Dear Dr. Carricart-Ganivet:

I'm pleased to inform you that your manuscript has been deemed suitable for publication in PLOS ONE. Congratulations! Your manuscript is now with our production department. 

Kind regards, 

on behalf of

Dr. Erik V. Thuesen 

Academic Editor

PLOS ONE